# The miRNA Content of Bone Marrow-Derived Extracellular Vesicles Contributes to Protein Pathway Alterations Involved in Ionising Radiation-Induced Bystander Responses

**DOI:** 10.3390/ijms24108607

**Published:** 2023-05-11

**Authors:** Ilona Barbara Csordás, Eric Andreas Rutten, Tünde Szatmári, Prabal Subedi, Lourdes Cruz-Garcia, Dávid Kis, Bálint Jezsó, Christine von Toerne, Martina Forgács, Géza Sáfrány, Soile Tapio, Christophe Badie, Katalin Lumniczky

**Affiliations:** 1Unit of Radiation Medicine, Department of Radiobiology and Radiohygiene, National Public Health Centre, 1097 Budapest, Hungary; csordas.ilona@nnk.gov.hu (I.B.C.); szatmari.tunde@nnk.gov.hu (T.S.); kisd@osski.hu (D.K.); forgacs.martina@nnk.gov.hu (M.F.); safrany.geza@nnk.gov.hu (G.S.); 2Doctoral School of Pathological Sciences, Semmelweis University, 1085 Budapest, Hungary; 3Centre for Radiation, Chemical and Environmental Hazards, UK Health Security Agency, Chilton, Didcot OX11 0RQ, UK; eric.andreasrutten@ukhsa.gov.uk (E.A.R.); lourdes.cruzgarcia@ukhsa.gov.uk (L.C.-G.); christophe.badie@ukhsa.gov.uk (C.B.); 4Helmholtz Zentrum München, German Research Center for Environmental Health GmbH (HMGU), 80939 München, Germany; psubedi@bfs.de (P.S.);; 5Federal Office for Radiation Protection (BfS), 85764 Oberschleissheim, Germany; 6Doctoral School of Biology, Institute of Biology, Eötvös Loránd University, 1053 Budapest, Hungary; jezso.balint@ttk.elte.hu; 7Research Centre for Natural Sciences, Institute of Enzymology, 1117 Budapest, Hungary; 8Metabolomics and Proteomics Core, Helmholtz Zentrum München, German Research Center for Environmental Health GmbH (HMGU), 80939 München, Germany; vontoerne@helmholtz-muenchen.de

**Keywords:** bone marrow, ionising radiation, extracellular vesicles, miRNA content, proteome, pathway analysis, bystander effects

## Abstract

Extracellular vesicles (EVs), through their cargo, are important mediators of bystander responses in the irradiated bone marrow (BM). MiRNAs carried by EVs can potentially alter cellular pathways in EV-recipient cells by regulating their protein content. Using the CBA/Ca mouse model, we characterised the miRNA content of BM-derived EVs from mice irradiated with 0.1 Gy or 3 Gy using an nCounter analysis system. We also analysed proteomic changes in BM cells either directly irradiated or treated with EVs derived from the BM of irradiated mice. Our aim was to identify key cellular processes in the EV-acceptor cells regulated by miRNAs. The irradiation of BM cells with 0.1 Gy led to protein alterations involved in oxidative stress and immune and inflammatory processes. Oxidative stress-related pathways were also present in BM cells treated with EVs isolated from 0.1 Gy-irradiated mice, indicating the propagation of oxidative stress in a bystander manner. The irradiation of BM cells with 3 Gy led to protein pathway alterations involved in the DNA damage response, metabolism, cell death and immune and inflammatory processes. The majority of these pathways were also altered in BM cells treated with EVs from mice irradiated with 3 Gy. Certain pathways (cell cycle, acute and chronic myeloid leukaemia) regulated by miRNAs differentially expressed in EVs isolated from mice irradiated with 3 Gy overlapped with protein pathway alterations in BM cells treated with 3 Gy EVs. Six miRNAs were involved in these common pathways interacting with 11 proteins, suggesting the involvement of miRNAs in the EV-mediated bystander processes. In conclusion, we characterised proteomic changes in directly irradiated and EV-treated BM cells, identified processes transmitted in a bystander manner and suggested miRNA and protein candidates potentially involved in the regulation of these bystander processes.

## 1. Introduction

Extracellular vesicles (EVs) are a heterogeneous class of nanoscale particles excreted by most cell types both under physiological and pathological conditions and under stress, and they serve primarily as intercellular communication vectors [1]. EVs include apoptotic bodies, microvesicles and exosomes, which are distinguished not primarily by size but by the method of biogenesis [2]. They shuttle several different kinds of cargo between cells, including lipids, proteins, DNA (DNA), mRNA, long non-coding RNA (lncRNA), and miRNA [3].

Apart from its cytotoxic and mutagenic effects, ionising radiation (IR) induces cellular stress and activates intercellular signalling mechanisms through which radiation damage can be transmitted in a bystander manner to cells not directly irradiated. This process is called the radiation-induced bystander effect, and it is a major mechanism through which tissue and systemic responses are elicited after local radiation damage [4,5,6]. Signals transmitted by chemokines, cytokines, small metabolites and various danger signal molecules released in the extracellular space play important roles in radiation-induced bystander responses. Given the complexity of bystander effects, it is less probable that single molecules mediate this response. EVs, with their complex cargo, are major candidates for mediating radiation-induced bystander responses, and several in vitro and in vivo studies actually proved the role of EVs in this process [7,8,9,10].

The role of bystander signalling is especially relevant in organs in which intercellular communication is a major element of proper organ functioning. Within the bone marrow (BM), interactions among the different stem and progenitor cells as well as the BM stroma are indispensable for normal haematopoiesis. In addition, BM is a particularly radiosensitive organ prone to the development of both IR-induced acute deterministic effects (acute BM damage) and late stochastic effects (radiation-induced leukaemia). EVs are known to play a major role in the communication with the BM microenvironment, being involved in the regulation of stem cell renewal, differentiation, proliferation and mobilisation [11,12].

Previously, we developed an in vivo model to study the role of EVs in mediating IR-induced bystander responses by injecting BM-derived EVs from irradiated into naïve, non-irradiated mice. We investigated local effects in the BM [9,10] and systemic effects in the blood and spleen [7,10,13]. In these studies, we showed that BM-derived EVs (1) induced changes in the pool of several of the BM haematopoietic and spleen immune cell subpopulations which mimicked direct irradiation effects; (2) induced DNA damage and apoptosis similarly to direct irradiation; (3) led to changes in several plasma protein levels involved in the inflammation and immune response, which were very similar to the effects observed after direct irradiation and (4) led to an increased oxidative stress. We also demonstrated that only EVs originating from the BM of acutely irradiated mice were able to initiate bystander responses, and these responses were long-lasting [9]. In the present study, we used a mouse model prone to radiation-induced leukaemia. We characterised IR-induced changes in the miRNA content of EVs and miRNA-regulated pathways as well as the proteome and related pathways of BM cells (BMC) treated with EVs from irradiated mice. By comparing pathways regulated by the differentially expressed miRNAs in the EVs with protein pathways altered in BMCs treated with EVs from irradiated mice, we identified miRNA candidates involved in bystander signalling in EV-recipient BMCs. The main steps of the workflow are presented in Figure 1.

To simplify the terminology within the manuscript, we will use the following terms: EVs will refer to BM-derived EVs, since BM was the only source of EV isolation in the present study. Control EVs will refer to EVs isolated from the BM of control, sham-irradiated mice and, consecutively, 0.1 Gy and 3 Gy EVs will refer to EVs isolated from the BM of mice irradiated with either 0.1 Gy or 3 Gy.

## 2. Results

BM-derived EVs were isolated from the BM supernatant of total body-irradiated CBA/CA mice and characterised as described [14]. The electron microscopic characterisation of EV morphology and size indicated typical EV structures, while EV-specific proteins were identified by Western blotting (Figure 2A,B). The average particle size measured with TRPS was 150 nm, with no significant differences between EVs from control or irradiated mice (Figure 2C).

### 2.1. The miRNA Cargo of BM-Derived EVs Shows a Dose-Dependent Response to IR

The total RNA extracted from control, 0.1 Gy and 3 Gy EVs from three independent experiments was run in an nCounter panel which included probes to identify 800 different murine miRNAs. Analysis performed with the DIANA Tools, mirExTra2.0 software indicated that the level of two miRNAs (mmu-miR-761 and mmu-miR-129-5p) was increased in 0.1 Gy EVs compared to in EVs from controls (Appendix A). Pathway analysis based on the KEGG database using the Diana Tools miR path software identified two significantly altered pathways (TGF-beta signalling pathway and Signalling pathways regulating pluripotency of stem cells) relevant for the radiation response in the BM (Appendix A).

In the 3 Gy EVs, 17 miRNAs were differentially expressed; two of them (mmu-miR-709 and mmu-miR-706) had decreased levels, while the others had increased levels (Appendix A). The two miRNAs with increased levels in 0.1 Gy EVs were also increased in 3 Gy EVs. Both miRNAs were present at higher levels in 3 Gy EVs than in 0.1 Gy EVs, suggesting dose-dependent alteration. No EV-derived miRNAs unique for low doses were identified. Pathway analysis revealed 34 significantly altered pathways potentially regulated by the 17 differentially expressed miRNAs in 3 Gy EVs (Appendix A). The majority of the pathways were linked to cancer (29%), signal transduction (26%) and cellular processes (15%). Pathways relevant for BM functioning (signalling pathways regulating pluripotency of stem cells, acute myeloid leukaemia, chronic myeloid leukaemia) were among the significantly altered pathways.

Clustering analysis performed with the BRB array tool showed clear up- and downregulated clusters of miRNAs, which also included the miRNAs identified using Diana tools software (Figure 3A,B). Dose correlation analysis identified 13 miRNAs that showed a strong dose-response (Table 1), with correlation coefficients between 0.95 and 0.84.

QRT-PCR measurements confirmed nCounter results for the downregulated miRNAs in the EVs identified by the DIANA Tools and mirExTra and also for most of the miRNAs correlating with the dose (based on BRB array analysis), though a dose–response relationship could not be identified for all miRNAs (Figure 4). Significant changes in miRNA levels were detected mostly after irradiation with 3 Gy (mmu-miR-669d-5p, mmu-miR-93-5p, mmu-miR-467h, mmu-miR-706, mmu-miR-1933-5p, mmu-miR-181c-5p, mmu-miR-338-5p), while 0.1 Gy induced significant changes only in two cases (mmu-miR-1933 and mmu-mi-34b-5p). Three miRNAs, which showed a positive dose correlation based on BRB array analysis, correlated negatively with the dose after qRT-PCR (mmu-miR-669d-5p, mmu-miR-504-5p and mmu-miR-467h). However, we have to point out that the nCounter system used the same ID for the identification of both mmu-miR-669d-5p and mmu-miR-467h, while individual miRNA-specific primers were used in the qRT-PCR reaction. Mmu-miR-34b-5p showed a positive dose correlation based on BRB array analysis, while qRT-PCR indicated an inverse dose relationship (decreased miRNA level after 0.1 Gy and enrichment after 3 Gy). For three miRNAs (miR-708-5p, mmu-miR-761 and miR-129-5p), we could not confirm any significant dose correlation with qRT-PCR (Figure 4).

### 2.2. Proteomic Changes in BMCs Treated with BM-Derived EVs from Irradiated Mice Indicate Partially Distinct Pathway Alterations to Directly Irradiated BMCs

Proteomic changes in BMCs treated with BM-derived EVs from irradiated mice have been compared to the proteome of directly irradiated BMCs. The ex vivo incubation of BMCs with fluorescently labelled EVs indicated that the rate of EV uptake was around 80%, with no significant difference in EV uptake if EVs originated from the BM of irradiated or control mice (Figure 5).

Proteomic changes were investigated in ex vivo cultured murine BMCs either non-irradiated (0 Gy or control), irradiated with 0.1 Gy or 3 Gy X-Rays or treated with BM-derived EVs isolated from mice irradiated with the same doses (0 Gy, 0.1 Gy or 3 Gy). A total of 3718 proteins were identified in the BMCs, among which 2707 proteins were qualified for further studies according to the selection criteria (FDR < 1% and identified with at least two unique peptides). Principal component analysis (PCA) and heat map clustering demonstrated that the 3 Gy group was the most different from all other experimental groups (Figure 6).

#### 2.2.1. Proteomic Changes in Directly Irradiated BMCs

Irradiation with 0.1 Gy led to 142 deregulated (33 downregulated and 109 upregulated) proteins (Appendix A, Figure 7A and Appendix A) compared to control samples. STRING clustering analysis indicated that the largest protein cluster included mitochondrial proteins involved in ATP synthesis and oxidative phosphorylation (Appendix A). Irradiation with 3 Gy led to 360 deregulated (236 downregulated and 124 upregulated) proteins in the BMCs (Appendix A, Figure 7A and Appendix A) compared to control samples, which could be grouped into several clusters. One big cluster contained proteins involved in biological processes related to DNA repair, replication and DNA metabolism, stress response and metabolic pathways. A further cluster contained proteins involved in chromatin remodelling and histone proteins. Mitochondrial proteins involved in electron transport, oxidative phosphorylation and oxidation-reduction complexes constituted a third cluster (Appendix A). Seventy-two proteins were in common between BMCs irradiated with 0.1 Gy and 3 Gy. Changes in the expression of 35 proteins correlated with the dose, with 50% or higher difference in the protein abundance ratio between 0.1 Gy and 3 Gy samples (Figure 7A and Appendix A; highlighted proteins indicate those correlating with the dose). Clustering analysis of the common proteins indicated that the largest protein clusters belonged to biological processes related to mitochondrial processes such as mitochondrial ATP synthesis and oxidative phosphorylation, as well as iron sulphur clusters (Figure 7B, Appendix A). The common proteins most significantly upregulated (highest abundance ratio and lowest *p* value) were putative transferase CAF17 homolog, mitochondrial (Iba57) and cysteine desulfurase mitochondrial (Nfs1) and also belonged to this cluster.

Based on pathway enrichment analysis using the PathDIP tool, deregulated proteins in BMCs irradiated with 0.1 Gy could be associated with 107 altered pathways (Appendix A and Figure 7C). The vast majority of the deregulated pathways were associated with immune processes (15%), signal transduction (11%) as well as cancer (9%) and metabolism (9%, including oxidative phosphorylation and pyruvate metabolism); nevertheless, cell growth and death-related pathways were also deregulated (Figure 7D). In the BMCs irradiated with 3 Gy, the 360 altered proteins could be associated with 87 altered pathways (Appendix A and Figure 7C). The highest number of altered pathways were related to metabolism (18%) and genetic information processing (17%). One-third of the pathways within genetic information processing were related to DNA damage and repair. Cancer-related pathways represented 10% of the total pathway alterations, including acute myeloid leukaemia. Immune-related and signal transduction pathways were also present but in lower percentages compared to BMCs irradiated with 0.1 Gy (Figure 7D).

Sixty-three pathways were in common between BMCs irradiated with 0.1 Gy and 3 Gy (Appendix A and Figure 7C). Common pathways were related to metabolism (14%), cancer (11%), genetic information processing (11%), signal transduction (10%), the immune system (8%) and cell growth and death (Figure 7E).

#### 2.2.2. Proteomic Changes in EV-Treated BMCs

The treatment of BMCs with 0.1 Gy EVs led to 106 deregulated (33 downregulated and 73 upregulated) proteins (Appendix A, Figure 7A and Appendix A) compared to BMCs treated with control EVs, which could be clustered into several biological functions, out of which mitochondria-related biological processes (ATP synthesis, electron transport, oxidation-reduction), stress response, signal transduction and immune processes represented the largest clusters (Appendix A). Based on pathway enrichment analysis, the 106 proteins could be associated with 15 pathways (Appendix A and Figure 7D), the majority of which were metabolism-related (Figure 7D). Treatment with 3 Gy EVs led to 63 deregulated (29 downregulated and 34 upregulated) proteins (Appendix A, Figure 7A and Appendix A) compared to BMCs treated with control EVs. STRING clustering analysis indicated fewer protein clusters than in the other treatment groups, the largest clusters belonging to nuclear and nucleosomal proteins involved in chromatin organisation, the regulation of histone methylation and DNA binding as well as spliceosomal complex assembly, mRNA splicing and ribonucleoprotein complex assembly. Protein clusters involved in immune processes and metabolic processes were also present (Appendix A). The 63 deregulated proteins could be associated with 117 pathways (Appendix A and Figure 7C). The most highly represented altered pathways were related to cancer (20%), signal transduction (16%) and the immune system (12%). Further altered pathways were endocrine system-related, genetic information processing including replication and repair and cellular processes including cell growth and death (Figure 7D).

Twenty-four proteins (Appendix A and Figure 7A) and four pathways (Appendix A and Figure 7C) were in common in BMCs treated with 0.1 Gy EVs and 3 Gy EVs, though common pathways could not be linked to either irradiation- or cancer- or BM damage-related processes (Figure 7E).

#### 2.2.3. Comparison of Proteomic Changes between Directly Irradiated and EV-Treated BMCs

Next, we identified deregulated common proteins and pathways between directly irradiated and EV-treated BMCs. Eighteen proteins were common in BMCs irradiated with 0.1 Gy and treated with 0.1 Gy EVs, out of which eight (Carbonyl reductase NADPH1 (Cbr1), Cytochrome b-c1 complex subunit 6, mitochondrial (Uqcrh), RUN and FYVE domain-containing protein 1 (Rufy1), N-acetylglucosamine-1-phosphodiester alpha-N-acetylglucosaminidase (Nagpa), Mitochondrial 2-oxodicarboxylate carrier (Slc25a21), Thioredoxin domain-containing protein 5 (Txndc5), Enoyl-[acyl-carrier-protein] reductase, mitochondrial (Mecr) and Mitochondrial import inner membrane translocase subunit Tim8A (Timm8a1)) could be associated with redox and mitochondrial processes (Appendix A and Figure 7A). Seven pathways were in common (Appendix A and Figure 7C), out of which oxidative phosphorylation was the most significant and also the only relevant for IR-related processes.

Twenty-six common proteins and 52 common pathways could be identified between BMCs directly irradiated and treated with 3 Gy EVs (Appendix A and Figure 7A). The 26 common proteins could not be grouped in big clusters; nevertheless, 3 common proteins (CD48 antigen (CD48), C-type lectin domain family 1 member B (Clec1b) and Integrin alpha-IIb (Itga2b)) were involved in platelet aggregation and activation, wound healing, cell adhesion and migration, 2 common proteins (p21-activated protein kinase-interacting protein 1 (Pak1ip1) and U4/U6 small nuclear ribonucleoprotein Prp31 (Prpf31)) were involved in spliceosomal complex assembly, mRNA splicing and ribonucleoprotein complex assembly, 2 proteins (Carbonyl reductase NADPH1 (Cbr1) and Carbonyl reductase NADPH2 (Cbr2)) were involved in NADPH activity and 2 other mitochondrial matrix proteins (Electron transfer flavoprotein-ubiquinone oxidoreductase, mitochondrial (Etfdh) and Propionyl-CoA carboxylase alpha chain, mitochondrial (Pcca)) were involved in propionyl-CoA carboxylase activity (Figure 7B). Pathway enrichment analysis indicated that out of the 52 common pathways, 17% were cancer-related, including acute myeloid leukaemia, 13% were signal transduction-related, 12% were related to genetic information processing, including replication and repair, 10% were linked to cell growth and death and 8% were linked to the immune system (Figure 7C,E).

### 2.3. Protein Pathways Regulated by EV-Containing miRNAs from Irradiated Mice Partially Overlap with Protein Pathways Altered in the EV-Treated BMCs

EV-derived miRNAs can potentially alter cellular pathways in EV-recipient cells by regulating their protein content. In order to identify those EV-derived miRNAs that are most probably involved in regulating cellular processes in the EV-recipient cells, we analysed overlaps in protein pathways predicted to be altered by EV-transmitted miRNAs and altered protein pathways based on proteomic data.

In 0.1 Gy EVs, two miRNAs had altered expression levels, which belonged to three pathways. BMCs treated with 0.1 Gy EVs led to 106 deregulated (up- or downregulated) proteins that could be linked to 15 pathways (Appendix A). The two pathway lists had no common elements; therefore, we assumed that EV-containing miRNAs after low-dose irradiation are not the major regulators of protein pathways in EV-acceptor BMCs.

The 17 miRNAs differentially expressed in 3 Gy EVs regulated 34 pathways. BMCs treated with 3 Gy EVs resulted in 63 (up- or downregulated) proteins which could be grouped in 117 pathways. Seventeen pathways were in common between miRNAs and proteins, and the vast majority of common pathways were cancer-related (59%) (Appendix A and Figure 8). These findings suggested that miRNAs from 3 Gy EVs might have important roles in regulating protein pathways in EV-acceptor BMCs.

In order to better characterise the interactions between EV-derived miRNAs and deregulated cellular proteins in the EV-acceptor BMCs, direct miRNA–protein interactions were analysed using the TARBASE database (for validated interactions) and the miRDB database (for predicted interactions). Out of the 17 miRNAs differentially expressed in the 3 Gy EVs, we identified those that could interact with at least one of the 63 deregulated proteins in the BMCs treated with 3 Gy EVs and found that 11 miRNAs could interact directly with 19 proteins. Nevertheless, only interactions between 6 miRNAs and 11 proteins were involved in the 17 altered pathways that were common between deregulated miRNAs in 3 Gy EVs and deregulated proteins in BMCs treated with 3 Gy EVs (Figure 9 and Appendix A). Alterations in the level of two highly significantly upregulated proteins ((Histone H2A.V (H2afv) and Cysteine protease ATG4B (Atg4b)) in the samples treated with 3 Gy EVs (for Atg4b *p* = 4.51 × 10^−16^ and for H2afv *p* = 3.63 × 10^−5^) and participating in validated interactions with miRNAs were investigated by Western blot analysis as well in order to confirm the proteomic data. The level of both proteins increased in the BMCs treated with 3 Gy EVs; nevertheless, changes were only significant for H2afv (Figure 10).

## 3. Discussion

In our previous studies, we reported that BM-derived EVs played an important role in transmitting IR-induced bystander effects in the BM [9,10,13]. Using the C57Bl/6 mouse model, we demonstrated that EVs isolated from the BM supernatant of irradiated mice, when injected systemically into naïve animals, induced changes in cell viability and cell subpopulation distribution in the BM and periphery that were similar to those for mice directly irradiated. We also highlighted the potential role of EV-derived miRNAs in mediating bystander effects. The aim of the present study was to identify those miRNAs in the BM-derived EVs of irradiated mice, which might interfere with cellular processes in the EV-acceptor cells. The CBA/CA mouse was used because this radiosensitive strain is prone to develop acute myeloid leukaemia exclusively upon irradiation [15].

Although blood is the most frequently used source of EVs for biological studies [16], blood-derived EVs are not optimal for characterising EV-mediated communication within a tissue microenvironment, since blood-derived EVs represent a mixture of EVs released from various parts of the body and, thus, they are not representative for a particular tissue microenvironment. Therefore, our study was performed using BM-derived EVs isolated from BM supernatant, in which EVs released by various BM cells are present in a concentrated form.

EVs carry a complex cargo comprising various proteins [17], different types of RNAs, various non-coding RNA types such as miRNAs [10,18], circular RNAs [19,20], long non-coding RNAs [21,22] as well as DNA of various sources such as genomic, mitochondrial [23], extrachromosomal [24], small metabolites [25] and possibly other biological structures present in the cells releasing the EVs. Out of these proteins and miRNAs are the best characterised molecules due to their role in regulating cellular processes in the EV-acceptor cells.

According to our results, the number of differentially expressed miRNAs in the EVs isolated from mice irradiated with low (0.1 Gy) or high (3 Gy) doses was low. In the EVs isolated from 0.1 Gy-irradiated mice, only two differentially expressed miRNAs were detected; nevertheless, they regulated important pathways involved in either signal transduction (TGF-β signalling pathway) or normal BM functioning (signalling pathways regulating the pluripotency of stem cells). Seventeen miRNAs were differentially expressed in EVs isolated from mice irradiated with 3 Gy, regulating mostly cancer-related pathways (including acute myeloid leukaemia) and multiple pathways relevant for either BM functioning (Wnt signalling pathway, signalling pathways regulating the pluripotency of stem cells) or for the cellular response to IR (Hippo signalling, cell cycle, TGF-β signalling, FoxO signalling, MAPK signalling, Erb signalling), suggesting the potential involvement of these miRNAs in mediating IR-induced bystander responses. The findings that all of the differentially expressed miRNAs of 0.1 Gy EVs were present in 3 Gy EVs and the expression level of several miRNAs in the EVs correlated with the applied dose suggest a dose-dependent regulation of miRNA packaging in EVs. Previous studies identified cellular and circulating miRNAs with various degrees of correlation with ionising radiation dose [26,27], but we are not aware of reports regarding the dose-dependent or dose-correlating expression of miRNAs in EVs.

Comparing our current results with the study of Szatmári et al., in which differentially expressed miRNAs in BM-derived EVs were investigated in C57Bl/6 mice irradiated with either 0.1 Gy or 2 Gy [10], only two miRNAs (miR709 and miR761) were in common in the EVs irradiated with high doses. Nevertheless, the number of common pathways regulated by the differentially expressed miRNAs in the two studies was largely overlapping. The two pathways regulated by miRNAs differentially expressed in the 0.1 Gy EVs (TGF-β signalling pathway and signalling pathways regulating the pluripotency of stem cells) and 80% of pathways regulated by the miRNAs from 3 Gy EVs were also present in the study by Szatmári et al.

We are not aware of any other publication investigating differentially expressed miRNAs in BM-derived EVs after IR. Comparing IR-induced differences in the miRNA content of EVs from different biological sources is difficult, since the miRNA content of the EVs largely reflects the miRNA panel of the EV-donor cells. Nevertheless, the majority of the EVs released by the BM should be present in the blood as well, although in a much lower yield. Therefore, we compared our current data with another study by Szatmári et al., in which they investigated differentially expressed miRNAs in EVs isolated from the blood of C57Bl/6 mice irradiated with either 0.1 Gy or 2 Gy [13]. Although identical miRNAs were not detected, several pathways relevant for cancer (such as acute myeloid leukaemia, chronic myeloid leukaemia, mTOR signalling) or signal transduction processes altered by IR (such as FoxO signalling, Hippo signalling, MAPK signalling, TGF-beta signalling) were in common. Moertl et al. investigated the effect of IR on the miRNA content of human peripheral blood mononuclear cell-derived EVs 72 h after irradiation with 2 Gy or 6 Gy. Similar to our data, they identified a limited number of differentially expressed miRNAs, and only the upregulated miRNA-338 was identical with the ones reported in our current study. MiRNA-regulated pathways related to the cell cycle, cell death and proliferation as well as DNA repair were also reported as altered [28]. The proteome of BMCs irradiated with 0.1 Gy revealed protein clusters involved in mitochondria-related processes, most notably ATP synthesis and oxidative phosphorylation, supporting that low-dose irradiation induces oxidative stress, as reported in other studies as well [29,30,31]. Pathway enrichment analysis indicated that immune system-related signalling pathways (such as NF-κB, T cell receptor, B cell receptor, Toll-like receptor, IL-17) and pathways related to immune cell differentiation and function were abundant, supporting the immune modulatory effect of low-dose irradiation, as reported [32,33,34,35].

The irradiation of the BMCs with 3 Gy led to a much higher number of deregulated proteins, clustered around biological processes relevant for cellular responses to high-dose radiation damage, such as DNA-related processes (repair, replication, DNA and nucleotide metabolism), stress response, chromatin remodelling, the regulation of transcription and mRNA maturation as well as several mitochondrial processes reflecting oxidative stress. Pathway enrichment analysis largely supported protein clustering, since the most significantly altered pathways were metabolism and genetic information processing, out of which one-third was related to DNA damage and repair (such as homologous recombination, DNA replication, mismatch repair, nucleotide excision repair). Metabolic pathways affected nucleotide metabolism, carbohydrate metabolism as well as oxidative phosphorylation. These data are in agreement with previous studies performed in various biological models indicating that high-dose irradiation leads to oxidative stress and the activation of DNA damage response pathways [36,37,38,39].

Approx. 50% of the common proteins between BMCs irradiated with 0.1 Gy and 3 Gy correlated with the dose, and these proteins clustered around the oxidative stress response, MAPK and NF-κB signalling and apoptosis, indicating that the severity of radiation-induced oxidative stress, inflammation and apoptosis is dose-dependent. Similar findings were reported by others as well [40,41,42]. Pathway enrichment analysis indicated 63 common pathways basically supporting protein clustering. Nevertheless, several important differences in pathway alterations between the two doses were also noted. Pathway alterations indicating radiation-induced DNA damage and pathways related to nucleotide metabolism were only present in BMCs irradiated with 3 Gy and were completely absent in low-dose-irradiated samples, indicating that low-dose irradiation does not activate DNA damage response pathways. These findings support previous reports with the same conclusion [43,44]. On the other hand, immune system-related and signal transduction pathways were less abundant in BMCs irradiated with high dose compared to low dose. While apoptosis was a common pathway for both doses, senescence-related pathways were only present after high-dose irradiation. Leukaemia-related pathway alterations were only present in BMCs irradiated with 3 Gy, which strongly supports previous data showing an increased incidence of myeloid leukaemia in the CBA/Ca mouse strain after irradiation with 3 Gy [15]. Altogether, commonalities and differences in the pathways between low- and high dose-irradiated BMCs highlight the major differences in the biological responses to the two doses. Systemic responses are markedly present after low-dose irradiation (increased number of immune system-related pathways), while mechanisms related to DNA damage repair (repair and replication-related pathways, metabolic pathways related to nucleotide metabolism) predominate in cellular responses to high-dose irradiation.

In order to study EV-induced changes in the protein profile of BMCs, first, we investigated whether the fact that EVs originate from irradiated mice influences their uptake rate by BMCs. Since we found no significant differences in the uptake rate, we concluded that changes in the protein pathways of BMCs treated with EVs were due to differences in the EV cargo and not due to differences in the number of internalised EV particles.

Proteomic changes in BMCs treated with 0.1 Gy EVs were clustered around mitochondrial processes involved in oxidative phosphorylation and were involved in 15 pathways, out of which 5 were metabolism-related such as oxidative phosphorylation and various components of lipid metabolism. The rest of the pathways were of less relevance for BM functioning and/or the response to IR. BMCs treated with 3 Gy EVs had 63 deregulated proteins that did not form big clusters except for a cluster of nuclear proteins involved in processes such as chromatin remodelling, histone methylation and DNA binding. Other smaller clusters were involved in immunological and metabolic processes. Pathway enrichment analysis indicated that, by far, the most highly represented group of pathways was cancer, followed by signal transduction and the immune response. Among cancer pathways, acute myeloid leukaemia- and chronic myeloid leukaemia-related pathways were significantly altered, indicating the potential role of EVs in leukemogenesis [45,46]. An additional important group of pathways was replication and repair, indicating that EVs could either transmit DNA damage into EV-recipient BMCs or signals leading to the activation of DNA damage response. This was also reflected in the alterations of certain signal transduction pathways interacting with the DNA damage response such as mTOR, PI3K-Akt, Wnt, cAMP, and NF-κB signalling pathways. Alterations in metabolism-related pathways were much less represented compared to either directly irradiated BMCs or BMCs treated with 0.1 Gy EVs. The comparison of common proteins and pathways between BMCs treated with 0.1 Gy EV and 3 Gy EV indicated that mostly proteins involved in energy metabolism and mitochondrial processes were in common; however, these were not reflected in identical pathway alterations.

Protein and pathway overlap between directly irradiated and EV-treated BMCs is a potential way to identify radiation-induced bystander signalling mechanisms transferred by EVs in the BM. As mentioned above, both direct irradiation with 0.1 Gy and the treatment of BMCs with 0.1 Gy EVs led to the deregulation of several mitochondrial proteins and protein pathways involved in energy metabolism. As such, 8 out of the 18 common deregulated proteins were involved in mitochondrial and redox-related processes, and the only common pathway that could be related to the IR response was oxidative phosphorylation. These data indicate that oxidative stress after the low-dose direct irradiation of BMCs can be transferred by EVs to non-irradiated BMCs in a bystander manner. EV-induced oxidative stress has been reported by other groups as well [47,48,49].

While we could not identify strong synergies between BMCs directly irradiated with 3 Gy or treated with 3 Gy EVs at the level of individual proteins, several altered pathways highly relevant for the IR response were identical between the two groups based on pathway enrichment analysis. Most importantly, all three pathways related to DNA replication and repair that were altered in BMCs treated with 3 Gy EVs were also altered in the directly irradiated cells, indicating that DNA damage response-related signals can be transmitted by EVs in a bystander manner. In addition, several of the signalling pathways related to DNA damage (PI3K-Akt, NF-κB) as well as cell death pathways characteristic of high-dose irradiation (apoptosis, senescence) were also in common between directly irradiated and EV-treated BMCs. The acute myeloid leukaemia pathway was present both in the directly irradiated and EV-treated BMCs. The number of cancer-related pathways was much higher in BMCs treated with 3 Gy EVs compared to their directly irradiated counterparts (23 vs. 9). Nevertheless, the majority of these pathways were related to cancer in specific organs other than BM, which might be due to signals originating from EVs present in the BM but released by cells from other organs, reaching the BM through the circulation.

Altogether, common protein and pathway alterations between directly irradiated and EV-treated BMCs indicated that cellular and molecular processes highly relevant for cellular responses to IR damage could be transmitted by EVs in a bystander manner. However, several important differences in the pathways could also be seen, as summarised in Appendix A.

Next, we were curious to see to what extent differentially expressed miRNAs of EVs isolated from irradiated mice were responsible for the transmission of bystander responses. Since the number of differentially expressed miRNAs in 0.1 Gy EVs was very low (only two miRNAs involved in three pathways, out of which one was unrelated to either IR or BM), we assumed that miRNAs from 0.1 Gy EVs do not play a major role in IR-induced bystander effects. On the other hand, the 17 differentially expressed miRNAs from 3 Gy EVs regulated several pathways relevant for IR-induced damage in the BM which overlapped with protein pathways identified in BMCs treated with 3 Gy EVs. Such common pathways with relevance in IR-induced BM damage were the cell cycle, pathways in cancer, transcription misregulation in cancer and acute and chronic myeloid leukaemia. Based on direct miRNA–protein interaction studies, 6 miRNAs and 11 proteins were identified that could directly interact and were involved in the mentioned pathways (Figure 9). This raises the possibility that alterations in the abovementioned protein pathways in the BMCs treated with 3 Gy EVs could be attributed to the regulatory role of the six EV-derived miRNAs. Further studies are needed to confirm the direct miRNA–protein interactions and their impact on the cell cycle and BM malignancies.

While several other pathways relevant for IR response were altered in the BMCs treated with 3 Gy EVs, the majority of these pathways could not be linked to specific miRNAs transmitted by 3 Gy EVs. This suggests that proteomic changes in BMCs treated with 3 Gy EVs might be only in small part regulated by the EV-derived miRNAs, highlighting the role of other molecules within the EVs able to transmit IR-induced bystander effects.

## 4. Materials and Methods

### 4.1. Mouse Model and Irradiation

CBA/Ca mice were used in all experiments. Mice were kept and investigated in accordance with the guidelines and all applicable sections of the 2011 CLVIII Hungarian law about animal protection and welfare and the European 2010/63/EU directives and regulations. All animal studies were approved, and permission was issued by the Budapest and Pest County Administration Office Food Chain Safety and Animal Health Board (ethical permission: PE/EA/392-7/2017).

Nine- to twelve-week-old male mice randomly selected from at least five different litters were either sham-irradiated (0 Gy, controls) or total-body-irradiated with 0.1 Gy and 3 Gy X-rays using an X-RAD 225/XLi X-ray source (Precision X-ray, North Branford, CT, USA). The mice were sacrificed 24 h after irradiation by using a 100 mg/kg (0.25 mL/kg) intraperitoneal (i.p.) injection of sodium pentobarbital.

### 4.2. Isolation of BMCs and Co-Culture of BM-Derived EVs with BMCs

Twenty-four hours after irradiation, the BMs were isolated from the femur and tibia of mice by flushing out the tissue from the diaphyses of the bones and suspended in PBS. A BM single-cell suspension was made by the mechanical disaggregation of the tissue; then, the cells were filtered by a 40 µm mesh filter. The cells were pelleted by centrifugation at 500× *g* and 4 °C for 10 min. The supernatant was removed and used for BM-derived EV isolation, as described below. Cell pellets were used for proteomic analysis, as described below.

BMCs from control (0 Gy or sham-irradiated) mice were prepared as described above and were incubated with EVs isolated from the BM supernatant of either control (0 Gy), 0.1 Gy- or 3 Gy-irradiated mice. Cell pellets were resuspended in RPMI1640 (Lonza Bioscience, Verviers, Belgium) containing 10% foetal bovine serum (FBS) (Euroclone S.p.a., Pero(MI), Italy). Live BMCs were determined by trypan blue exclusion, and 20 × 10^6^ BMCs per sample were incubated with EVs isolated from the supernatant of 100 × 10^6^ BMCs for 3 h at 37 °C and 5% CO_2_. BMCs from directly irradiated mice were processed in a similar manner, without adding EVs. After the three-hour incubation time, 20 × 10^6^ cells per sample were pelleted at 500× *g* and 4 °C, and cell pellets were snap-frozen in liquid nitrogen and kept at −70 °C until proteomic analysis.

Part of the EVs and BMCs were used to trace the EV uptake. EVs were isolated from the BM supernatant of 0 Gy-, 0.1 Gy- or 3 Gy-irradiated mice, as described above. Isolated EVs were stained with a cell permeant nucleic acid stain that is selective for RNA (SYTO^®^ RNASelect™ Green Fluorescent Cell Stain Invitrogen, Waltham, MA, USA), following the manufacturer’s protocol. Briefly, EVs were incubated with the dye (1 µM final concentration) at 37 °C in the dark for 20 min, and the excess of nonincorporated dye was removed using a GE Healthcare PD SpinTrap G-25 desalting column (GE Healthcare, Life Science, Chicago, IL, USA). Stained EVs were incubated with freshly isolated BMCs at a 1/5 ratio for 3 h at 37 °C; the non-internalised EVs were removed by centrifugation at 400× *g* for 10 min. As a control, PBS was stained and incubated with BMCs in the same way. The EV uptake was assessed by measuring the acquired fluorescence with a CytoFlex flow cytometer (Beckman Coulter, Brea, CA, USA). The results were analysed using the Cytexpert software version 2.3.0.84 (Beckman Coulter, Brea, CA, USA).

### 4.3. Isolation, Validation and Quantification of Mouse BM-Derived EVs

BM-derived EVs were isolated from the BMC supernatant of control and irradiated mice by pooling the supernatants of five mice/irradiation doses/experiments. EV isolation was conducted 24 h after irradiation using the ExoQuick-TC kit (System Biosciences, Palo Alto CA, USA), as described previously [10]. EVs were used for miRNA analysis and for co-culture with BMCs.

BM-derived EVs were investigated by transmission electron microscopy (TEM) as described [14]. The size distribution analysis of EVs was conducted by tunable resistance pulse sensing (TRPS) using an NP150 nanopore (measurement range: 70–420 nm) [9]. EV-specific protein markers were investigated by Western blot analysis, as detailed below.

### 4.4. RNA Extraction from BM-Derived EVs for miRNA Analysis

Total RNA extraction from BM-derived EVs was performed using the RNeasy mini extraction kit (Qiagen, Hilden, Germany), according to the manufacturer’s instructions. The MiRNA concentration was measured on a BioAnalyzer 2100 (Agilent, Santa Clara, CA, USA) according to the manufacturer’s instructions. Briefly, the concentrations of the samples to be measured were standardised according to the total RNA concentration, as measured by NanoDrop (ThermoFisher, Waltham, MA, USA), between 20 ng/µL and 50 ng/µL depending on the available sample amount. These were then loaded onto the BioAnalyzer small RNA chip (Agilent, part number 5067-1548) and read by the BioAnalyzer 2100. The resultant miRNA concentration was calculated by multiplying the measured miRNA concentration by the dilution factor of the sample, yielding the miRNA concentration in the undiluted sample.

### 4.5. nCounter Analysis of BM-Derived EV miRNAs and Their Quantitative Validation by Real-Time Polymerase Chain Reaction (RT-PCR)

An nCounter Mouse v1.5 miRNA Expression Panel including 800 miRNAs (NanoString Technologies, Seattle, WA, USA) was used to analyse the levels of miRNA expression in EVs, according to the manufacturer’s instructions. MiRNA expression was analysed in four biological replicates in each experimental group (0 Gy, 0.1 Gy and 3 Gy EV, respectively). Briefly, the miRNA samples prepared as described above were hybridised to a miRNA-specific probe, which in turn hybridises to a barcoded fluorescent reporter specific to that probe; the barcode identifies which miRNA it is attached to. These were then fixed to a chip, which was read by the nCounter machine, giving counts based on recorded fluorescence intensities. This allows for a molecule-by-molecule resolution of the relative abundance of the miRNA in a sample.

The nCounter-reported raw counts were normalised according to the top 100 miRNAs using nSolver 4.0 software, and the normalised data were then analysed with BRBArray Tools, an open-source analysis extension used by the National Cancer Institute [50]. Dose correlation analysis and heat maps were also generated using BRBArray Tools. Statistical relevance was tested with DIANA Tools, mirExTra2.0. [51], using the LIMMA statistical method. The *p*-value threshold for relevance was set at 0.05.

cDNA synthesis and miRNA analysis by RT-qPCR were carried out by using the miRCURY™ LNA™ miRNA PCR System (Qiagen). cDNA was prepared from 2 µL RNA with a concentration of 40 ng/µL in a 10 μL reaction volume. The cDNA was diluted 20-fold and assayed in a 10 μL PCR reaction volume. The amplification was performed in a Rotor-Gene Q real-time PCR cycler (Qiagen). The following PCR primers (all purchased from Qiagen) were used: miR-669d-5p (YP00205051), miR-93-5p (YP00204715), miR-504-5p (YP00204396), miR-467h (YP00205922), miR-706 (YP00205976), miR-709(YP00205463), miR-1961 (YP00205381), miR-1933-5p (YP00205351), miR-181c-5p (YP00204683), miR-338-5p (YP00204114), miR-34b-5p (YP00205075), miR-761 (YP00205475), miR-129-5p (YP00204534) and miR-708-5p (YP00204490). The amplification curves were analysed using the Rotor-Gene Q Series software (software version 2.1.0.9) both for the determination of quantification cycles and for the melting curve (T_m_) analysis. In order for the data to be considered for further analysis, the following criteria had to be met: appropriate melting curves, T_m_ had to be within known specifications for the assay and the Cq value had to be ≤37. The relative concentration of each miRNA was calculated by the Rotor-Gene Q software, where sham-irradiated (0 Gy) samples were used as controls. To achieve optimal relative expression results, mmu-miR-423-3p was used as a normaliser, since mmu-miR-423-3p was present in a constant and well-detectable concentration in BM-derived EVs, which did not change due to 0.1 Gy or 3 Gy irradiation.

### 4.6. Western Blot Analysis of EV and BMC Proteins

BMCs and BM-derived EVs were isolated as described above. After the EV isolation protein content was measured by the Bradford protein assay kit (Thermo Fisher Scientific, Waltham, MA, USA) using a Synergy HT (Biotek, Winooski, VT, USA) plate reader, EVs with a 60 µg protein content and 2.6 × 10^6^ BMCs were lysed with radioimmunoprecipiation assay (RIPA) lysis buffer containing 2% protease inhibitor (Protease Inhibitor Cocktail (P8340), Kenilworth, NJ, USA); then, the protein concentration was determined by the bichinchoninic acid (BCA) assay kit (Pierce™ BCA Protein Assay Kit, Thermo Fisher Scientific, Waltham, MA, USA). For further steps, 40 µg protein was precipitated by trichloroacetic acid (TCA) solution (one part of the TCA was added to three parts of the protein solution), incubated on ice for 5 min and pelleted at 3800× *g* and 4 °C for 5 min. The pellet was washed with ice-cold acetone twice, and the protein samples were diluted 1:2 in 2× Laemmli buffer (Bio-Rad Hercules, CA, USA) supplemented with β-mercaptoetanol, boiled at 95 °C for 5 min and cooled on ice for 5 min before loading on the gel. A total of 40 µg of the protein was loaded and electrophoresed on 4–20% sodium dodecyl sulphate-polyacrylamide (SDS-PAGE) gel (4–20% Mini-PROTEAN^®^ TGX™ Precast Protein Gels, Bio-Rad Hercules, CA, USA) and transferred to a polyvinylidene fluoride (PVDF) membrane. After blotting, the PVDF membrane was blocked with blocking buffer (containing 3% bovine serum albumin (BSA) in Tris Buffered Saline, with Tween 20) at room temperature (RT) for 30 min. The blocked membrane was incubated with the primary antibodies: recombinant Anti-ATG4B (ab154843), H2AFV Polyclonal Antibody (PA5109802) and beta Actin Polyclonal Antibody (PA1-183) (Thermo Fisher Scientific, Waltham, MA, USA) at RT for 2 h, followed by 1 h of incubation with horseradish peroxidase-conjugated goat anti-rabbit secondary antibody (Goat Anti-Rabbit IgG H&L (HRP), (ab6721 Abcam). Antibodies were diluted in the blocking buffer according to the manufacturer’s instructions. As a protein standard, the Spectra™ Multicolor Broad Range Protein Ladder was used (Thermo Fisher Scientific). The membrane was washed in Tris-buffered saline-tween buffer three times, and protein bands were visualised using 3,3′-diaminobenzidine substrate (Pierce™ DAB Substrate Kit, Thermo Fisher Scientific) by the chromogenic method. The density of each protein band was recorded and analysed by ImageJ software (Image Processing and Analysis in Java, National Institutes of Health, Bethesda, MD, USA). The measured density values were normalised to the density value of the loading control, β-actin.

### 4.7. Mass Spectrometry (MS) Sample Preparation and Measurement

BMCs from directly irradiated mice (n = 3 for 0 Gy BM, n = 4 for 0.1 Gy and 3 Gy BM) as well as BMCs co-cultured with BM-derived EVs from irradiated mice (n = 4 for 0 Gy BM + EV, n = 5 for 0.1 Gy BM + EV, 3 Gy BM + EV) were placed in 100 µL RIPA buffer (Thermo Fisher Scientific) that contained 25 mM Tris.HCl ph 7.6, 150 mM NaCl, 1% NP-40, 1% sodium deoxycholate and 0.1% SDS and incubated at 4 °C for 30 min [52]. They were then subjected to an ice-cold sonication bath for 30 s and another incubation (4 °C, 15 min). The protein concentration of the individual samples was determined using a BCA assay following the instruction manual (Thermo Fisher Scientific) on an Infinite M200 Spectrophotometer (Tecan GmbH, Crailsheim, Germany). BSA was used as an internal standard.

A total of 10 µg of the sample was enzymatically digested using a modified filter-aided sample preparation (FASP) protocol, as described in [53,54]. Peptides were stored at −20 °C until the MS measurement.

The MS measurement was performed in the data-dependent (DDA) mode. MS data were acquired on a Q Exactive (QE) high-field (HF) mass spectrometer (Thermo Fisher Scientific Inc.), as described in [55].

### 4.8. MS Data Processing and Protein Identification

Proteome Discoverer 2.4 software (Thermo Fisher Scientific; version 2.4.1.15) was used for peptide and protein identification via a database search (Sequest HT search engine) against the Swissprot mouse database (Release 2020_02, 17061 sequences), considering full tryptic specificity and allowing for up to one missed tryptic cleavage site, a precursor mass tolerance of 10 ppm and a fragment mass tolerance of 0.02 Da. Carbamidomethylation of cysteine was set as a static modification. Dynamic modifications included the deamidation of asparagine and glutamine, the oxidation of methionine and a combination of methionine loss with acetylation on protein N-terminus. The percolator was used for validating peptide spectrum matches and peptides, accepting only the top-scoring hit for each spectrum and satisfying the cut-off values for FDR < 1% and a posterior error probability < 0.01. The final list of proteins complied with the strict parsimony principle.

A schematic illustration of the workflow used for MS data processing and protein identification can be seen in Appendix A.

### 4.9. Data Processing and Label-Free Quantification

The quantification of proteins was based on the area value of the abundance values for unique plus razor peptides. Abundance values were normalised in a retention time-dependent manner to account for sample loading errors. The protein abundances were calculated by summing up the abundance values for admissible peptides. The final protein ratio was calculated using the median abundance values of three replicate analyses each. The statistical significance of the ratio change was ascertained by employing the *t*-test approach described in [56], which is based on the presumption that we look for expression changes for proteins that are just a few in number in comparison to the number of total proteins being quantified. The quantification variability of the non-changing “background” proteins can be used to infer which proteins change their expression in a statistically significant manner.

The T-test was solely used for pairwise comparisons of two conditions. The following pairwise conditions were compared: (0.1 Gy BM)/(0 Gy BM); (3 Gy BM)/(0 Gy BM); (0.1 Gy BM + EV)/(0 Gy BM + EV); (3 Gy BM + EV)/(0 Gy BM + EV).

Proteins identified with an FDR <1% and a fold change above ±1.33 in treated samples, along with a Benjamini–Hochberg adjusted *p*-value < 0.05, were considered deregulated.

### 4.10. Pathway Analysis of Significantly Altered EV-Derived miRNAs and BMC Proteins

Statistically significant miRNAs differentially expressed in the EVs from irradiated mice were used in the miRNA functional KEGG pathway analysis, which was performed with DIANA mirPath v.3. Experimentally validated interactions from TarBase v.8 (http://www.microrna.gr/tarbase (accessed on 20 September 2022)) [57] were used primarily, and data generated by the microT-CDS algorithm [58] were used only if no experimentally validated interaction was found for the miRNA. The *p*-value threshold was set at <0.05.

For protein analysis, all four treatment groups (0.1 Gy or 3 Gy directly irradiated BMCs, and BMCs treated with 0.1 Gy- or 3 Gy-irradiated EVs) were considered separately, and the results were compared to their respective controls (either 0 Gy sham-irradiated or treated with BM-derived EVs from 0 Gy-irradiated mice) and to each other. Only significantly deregulated proteins based on MS analysis were included in the analysis, where the official gene symbol of the proteins was used in the query data. To calculate the number of protein–protein interactions (PPI) and form protein clusters, the STRING database version 11.0 (https://string-db.org/ (accessed on 12 September 2022)) was used. In the analysis, all interaction sources (text mining, experiments, databases, co-expression, neighbourhood, gene fusion, co-occurrence) were used, and the minimum required interaction score was 0.4 (medium confidence); protein clustering was performed using the MCL clustering algorithm within STRING. Pathway annotations and enrichment analysis were performed with the pathDIP database, version 4.0.7.0 (http://ophid.utoronto.ca/pathDIP/ (accessed on 19 September 2022)), which is an annotated database integrating data from several pathway databases. The advantage of using PathDIP was that it gave a greater protein coverage by using both literature-curated (core), orthologue and extended pathways. Extended pathways integrate core pathways with experimentally detected and orthologous PPIs. PPIs are either experimentally detected or high-confidence computationally predicted. Due to the pathway extension method, more than 36,000 pathway orphans (proteins with no annotations available in curated or orthologue pathways) could be annotated in sixteen non-human organisms, which led to a 9.56 times increase in the protein coverage in model organisms [59]. In our PathDIP analysis, we only considered data originating from the KEGG database in order to use similar conditions as in the microRNA pathway analysis. We used extended pathway associations, where core pathways were integrated with experimentally detected and orthologous PPIs with a minimum confidence level of protein–pathway association predictions of 0.95. Only significant pathways (*p* ˂ 0.05) were considered.

To detect possible interactions between the significantly altered proteins and microRNAs, two databases were used: Tarbase for experimentally validated interactions and miRDB (http://mirdb.org/ (accessed on 20 September 2022)) [60] for predicted interactions.

### 4.11. Statistical Analysis

Data are presented as the mean ± standard deviation (SD). If no other method was indicated, than Students’s *t*-test was applied to determine statistical significance using GraphPad Prism version 6.00 for Windows (GraphPad Software, La Jolla, CA, USA). Data were considered statistically significant if the *p*-value was lower than 0.05. For the dose correlation calculation, Pearson Correlation was used. To identify significantly altered miRNAs, the LIMMA method was used in DIANA mirExTra 2.0.

## 5. Conclusions

In this study, we identified differentially expressed miRNAs in the BM-derived EVs from mice irradiated with low- or high-dose IR. We showed that miRNAs regulated pathways related to the IR response and to normal and malignant BM functioning. The miRNA profile of the EVs was only moderately affected by low-dose irradiation, but the deregulated miRNAs were involved in pathways that are important for a healthy bone marrow, such as signalling pathways regulating the pluripotency of stem cells and mitochondria-related biological processes. These data indicate that oxidative stress after low-dose irradiation can be transferred by EV-derived miRNAs to non-irradiated bone marrow cells in a bystander manner.

We performed detailed proteomic analysis of either directly irradiated BMCs or those treated with EVs from irradiated mice. Several proteins, protein clusters and protein pathways were in common between directly irradiated and EV-treated BMCs, most probably representing biological processes transmitted via EVs in a bystander manner. One such important common process was the induction of oxidative stress after low-dose direct irradiation and treatment with EVs from low-dose irradiated mice. While pathways related to the DNA damage response were altered in both high-dose-irradiated BMCs and in BMCs treated with 3 Gy EVs, there was an important difference at the protein level. In directly irradiated cells, protein clusters indicated alterations in processes directly involved in DNA damage recognition, repair and processes linked to the epigenetic control of the IR response such as chromatin remodelling or histone methylation. In the EV-treated cells, only protein clusters involved in biological processes related to chromatin remodelling and histone modifications were present, which we think indicates that EVs can transmit factors influencing the DNA damage response pathway only via epigenetic mechanisms. We are not aware of other studies comparing direct and EV-induced changes in the proteomic profile of BMCs after irradiation. Since EV-derived miRNAs have important roles in regulating cellular processes in EV acceptor cells, pathways regulated by differentially expressed miRNAs in the EVs and altered protein pathways in BMCs treated with BM-derived EVs were compared. Certain important pathways altered in BMCs treated with 3 Gy EVs were identified as also regulated by miRNAs differentially expressed in 3 Gy EVs such as the cell cycle and myeloid leukaemia. Within these common pathways, direct interactions were shown between six miRNAs (mmu-miR-706, mmu-miR-709, mmu-miR-761, mmu-miR-669g, mmu-miR-129-5p and mmu-miR-1942) and eleven proteins (Atg4b, Calu, H2afv, Krt16, Rtf1, Snca, Tgm2, Ube2c, Clic4, Dnajb6 and Eif2b1), which might represent potential biomarkers indicative of EV-mediated effects. The existence of such common pathways indicates that miRNAs can induce functional changes in EV acceptor cells, strengthening the possible regulatory role of EV-derived miRNAs. Nevertheless, the majority of altered protein pathways could not be linked with miRNA-regulated pathways, indicating that miRNAs are involved in the regulation of certain but not all EV-mediated bystander effects, and other molecules comprising the EV cargo also have their role. In order to understand IR-induced bystander signals mediated by EVs, it is important to perform the complex characterisation of EV cargo changes after IR and to link them to biological processes in the EV acceptor cells. Since bystander responses are very important modulators of radiation damage, a better knowledge of these processes helps in an improved understanding of tissue responses to IR and in the estimation of long-term risks after radiation exposure.

## Figures and Tables

**Figure 1 ijms-24-08607-f001:**
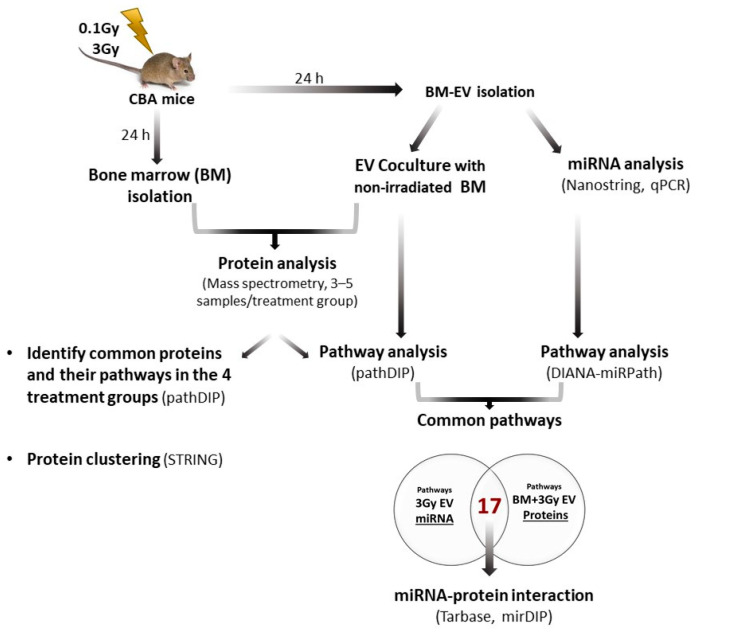
Schematic figure of the workflow. EV characterization is shown in Figure 2, EV uptake upon EV and BMC coculture is shown in Figure 5. Results of the miRNA expression analysis are shown in Figures 3 and 4 and Table 1. Summary of mass spectrometric results are presented in Figure 6. Results of miRNA and protein pathway analysis alongside with protein clustering results are shown in Figure 7. miRNA and protein interaction results are presented in Figures 8–10.

**Figure 2 ijms-24-08607-f002:**
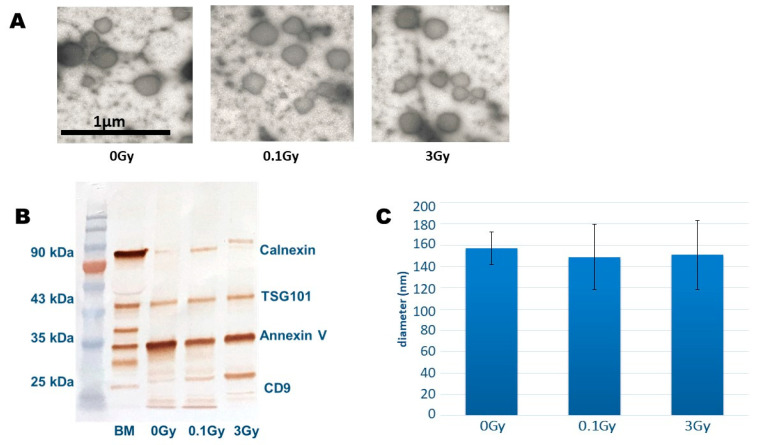
Characterisation of Bone Marrow-Derived Extracellular Vesicles. (**A**) Representative transmission electron microscopy images of extracellular vesicles isolated from the bone marrow of mice irradiated with the indicated doses of ionising radiation. (**B**) Representative Western blot analysis of whole cell lysates and extracellular vesicles isolated from the bone marrow of mice irradiated with the indicated doses of ionising radiation. Lane 1: protein ladder, lane 2: bone marrow whole cell lysate, lane 3–5: extracellular vesicle sample from control, nonirradiated, 0.1 Gy- and 3 Gy-irradiated mice. (**C**) Size of extracellular vesicle suspensions was examined by tunable resistance pulse sensing using an NP150 nanopore (measurement range: 70–420 nm). Mean values of extracellular vesicle size are shown, with bars representing standard deviations (SD). n = 3.

**Figure 3 ijms-24-08607-f003:**
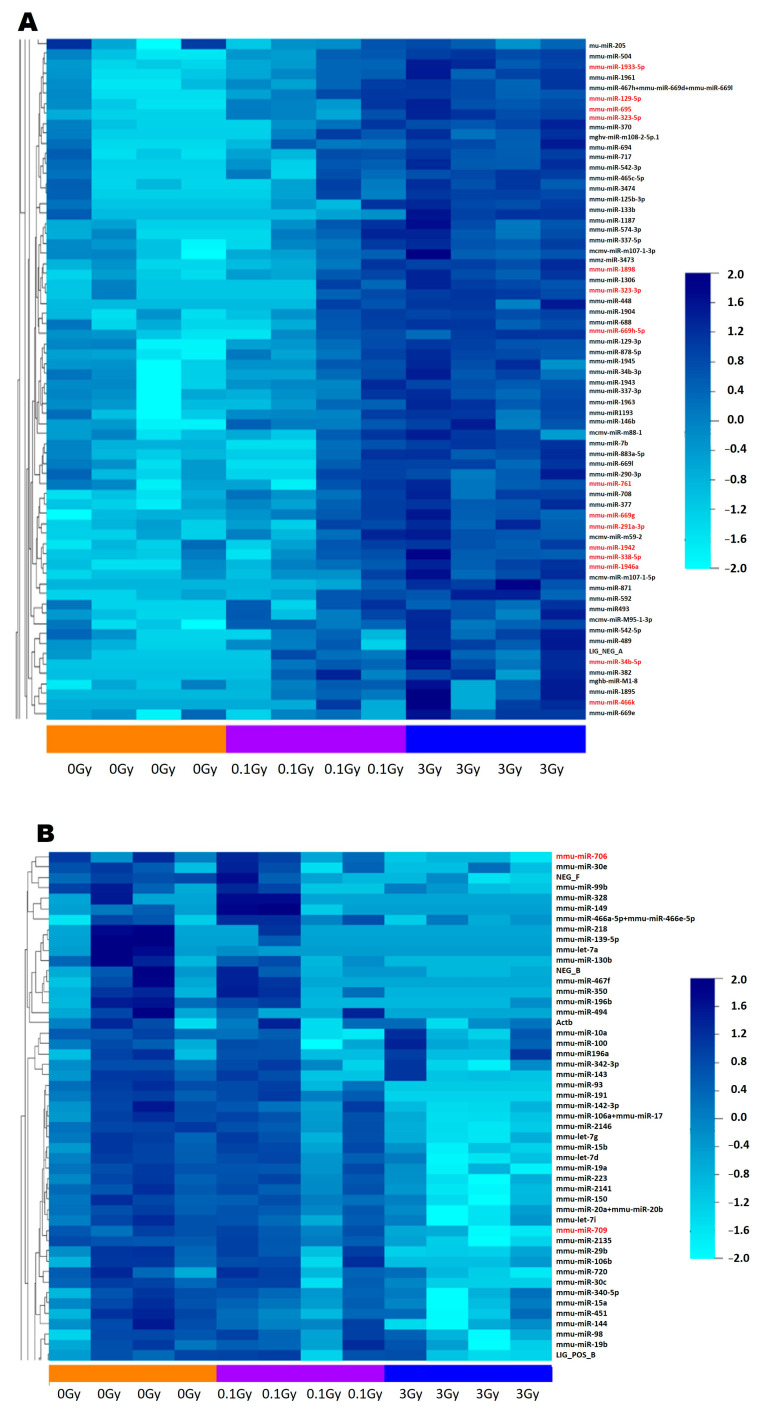
Based on clustering and dose correlation analysis, miRNAs from the bone marrow-derived extracellular vesicles of mice irradiated with 0.1 Gy and 3 Gy correlate with the dose. Heatmap and cluster dendrogram showing (**A**) a cluster of upregulated miRNAs and (**B**) a cluster of downregulated miRNAs from bone marrow extracellular vesicles of mice irradiated with 0.1 Gy and 3 Gy. MiRNAs highlighted in red are those miRNAs that were identified by Diana tools mirExTra 2.0. as significantly up- or downregulated. Clustering and dose correlation analysis was performed with the BRB array tools.

**Figure 4 ijms-24-08607-f004:**
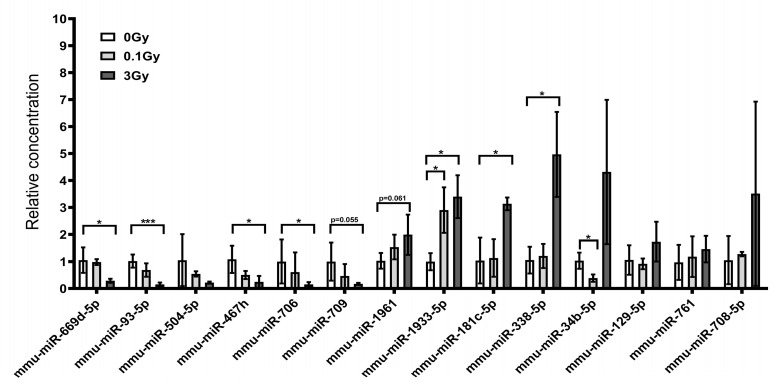
Ionising radiation influences the miRNA content of bone marrow-derived extracellular vesicles in irradiated mice. Extracellular vesicles were isolated from the bone marrow of control and irradiated mice, miRNAs were purified from the extracellular vesicles and the relative concentration of miRNAs was measured by qRT-PCR, as described in the Materials and Methods section. n = 3; Significant changes are indicated with * *p* ˂ 0.05 and *** *p* < 0.001.

**Figure 5 ijms-24-08607-f005:**
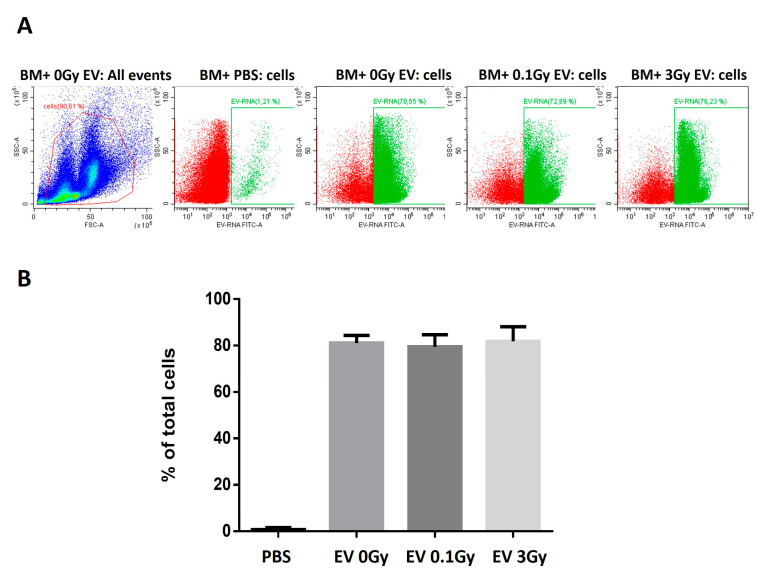
Irradiation does not influence the extracellular vesicle uptake by bone marrow cells. Bone marrow cells from control, non-irradiated mice were co-cultured in vitro with fluorescently labelled bone marrow-derived extracellular vesicles isolated from control, 0.1 Gy- and 3 Gy-irradiated mice, and the uptake rate was determined by flow cytometry, as described in Materials and Methods. (**A**) Flow cytometry blots showing the representative uptake rates of extracellular vesicles by bone marrow cells. Green colour indicates cells with EV uptake, red colour indicates cells without EV uptake. (**B**) The percentage of bone marrow cells taking up extracellular vesicles. n = 3. Error bars represent standard deviation (SD).

**Figure 6 ijms-24-08607-f006:**
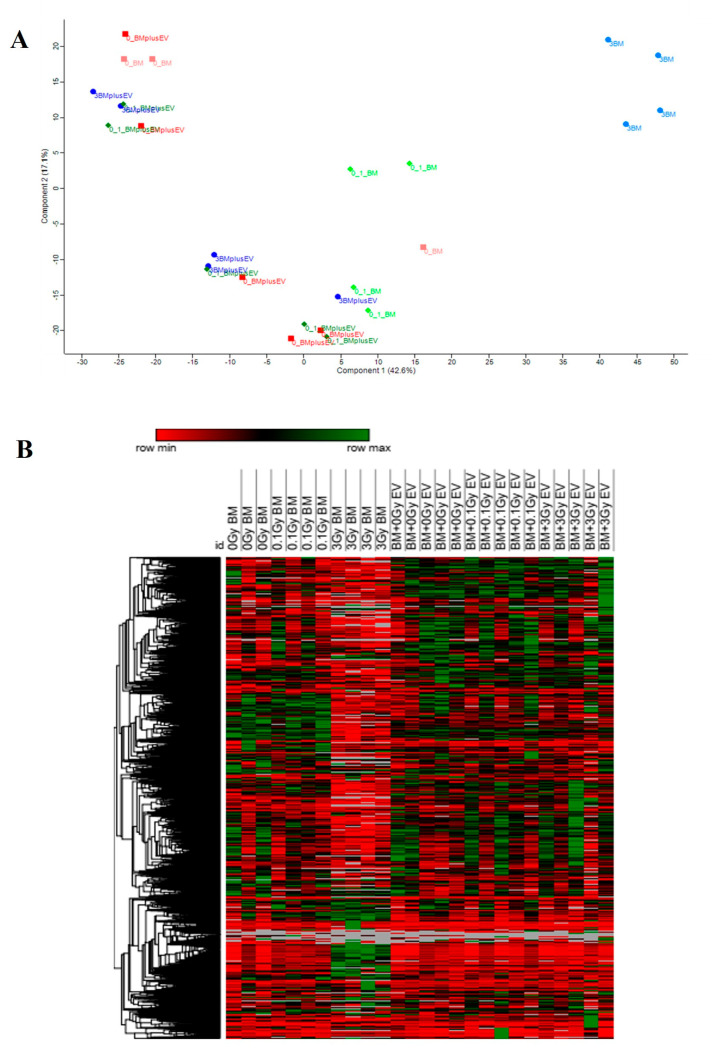
Irradiation of bone marrow cells with 3 Gy leads to a distinct proteomic pattern. (**A**) Principal component analysis was performed in Perseus software version 2.0.7.0. Proteome discoverer output with all proteins identified in all samples with more than two peptides was loaded. After log2 transformation, missing values were low-abundance-imputed according to program settings. Categorical annotations were added for experimental groups. Principal component analysis was performed using the standard settings of Perseus, displaying experimental groups. (**B**) Heatmap clustering. Proteins were characterised by hierarchical clustering (Euclidean distance algorithm and average distance method).

**Figure 7 ijms-24-08607-f007:**
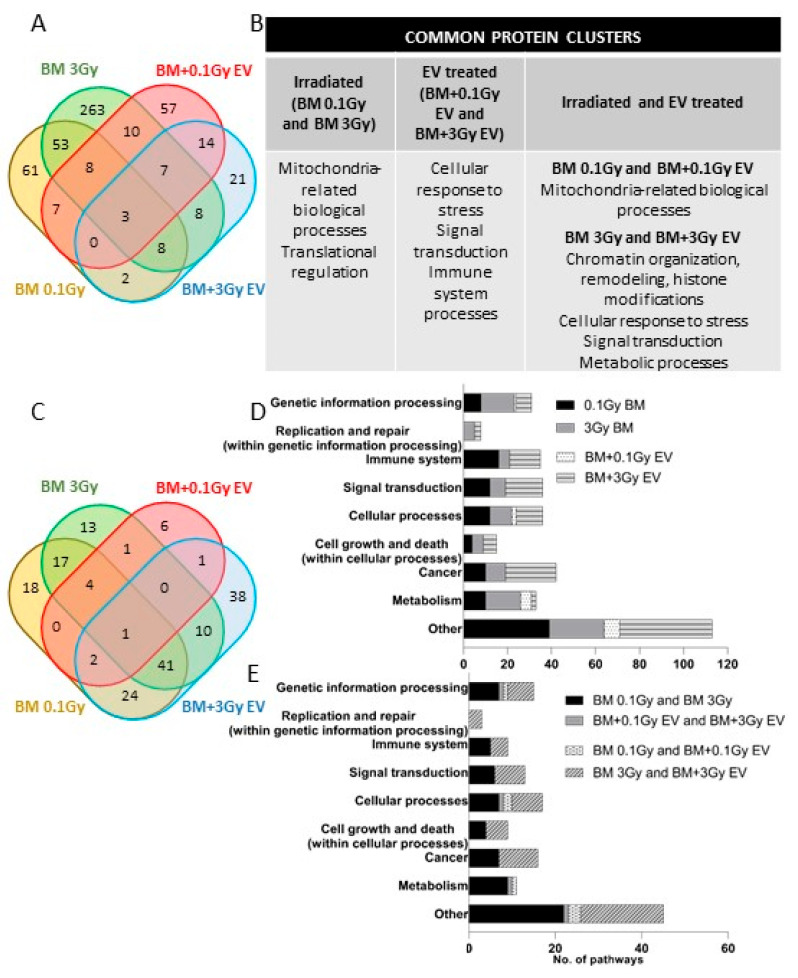
Significantly deregulated proteins and associated pathways in directly irradiated and extracellular vesicles-treated bone marrow cells. (**A**) Venn Diagram analysis of significantly deregulated proteins identified across all four treatment groups. (**B**) Common clusters formed by significantly deregulated proteins, as performed by STRING clustering analysis. (**C**) Venn diagram analysis of significant protein pathways. (**D**) Pathway distribution in the different treatment groups. (**E**) Pathway comparison between treatment groups. Pathway annotations and enrichment analysis were performed with the pathDIP database, integrating data from the KEGG database. BM: bone marrow; EV: extracellular vesicles.

**Figure 8 ijms-24-08607-f008:**
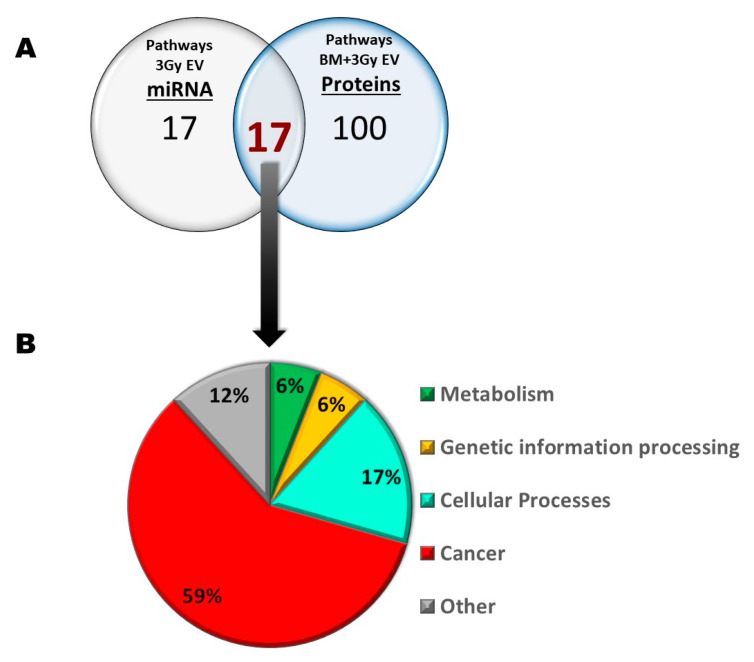
Common pathways between differentially expressed miRNAs from bone marrow-derived extracellular vesicles from mice irradiated with 3 Gy and deregulated proteins in bone marrow cells treated with bone marrow-derived extracellular vesicles of mice irradiated with 3 Gy. (**A**) Number of pathways; (**B**) Distribution of KEGG pathway classes.

**Figure 9 ijms-24-08607-f009:**
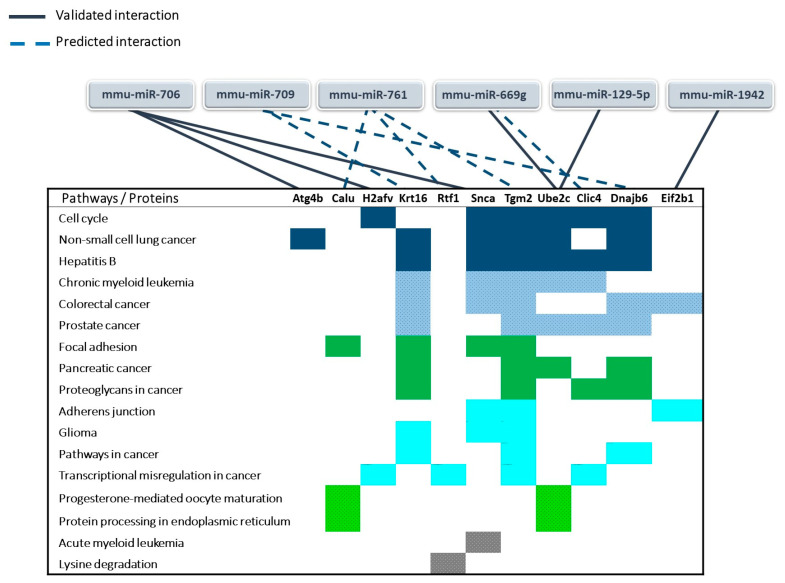
MiRNA–protein interactions involved in the common pathways between differentially expressed miRNAs from bone marrow-derived extracellular vesicles from mice irradiated with 3 Gy and deregulated proteins in bone marrow cells treated with bone marrow-derived extracellular vesicles of mice irradiated with 3 Gy. Only those miRNA and protein interactions are presented where both the miRNA and the protein could be linked to certain common pathways. MiRNA–protein interactions were searched for in the Tarbase (validated interactions) and miRDB databases (predicted interactions). Colour blocks indicate an identical number of proteins involved in a particular pathway.

**Figure 10 ijms-24-08607-f010:**
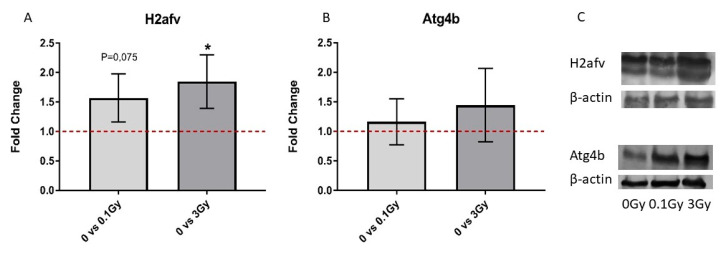
Western blot validation of deregulated proteins in bone marrow cells treated with bone marrow-derived extracellular vesicles from irradiated mice. A total of 40 μg of protein was loaded on the gel and hybridised to either anti-Atg4b or anti-H2afv or β-actin (loading control), as described in Materials and Methods. Quantification was performed with ImageJ software. Relative fold changes compared to bone marrow cells treated with extracellular vesicles from non-irradiated mice are shown in (**A**) for H2afv and (**B**) for Atg4b. (**C**) Representative blot image. n = 3. Error bars represent standard deviations (SD). Significant changes are indicated with * *p* ˂ 0.05.

**Table 1 ijms-24-08607-t001:** MiRNAs correlating with the dose. Dose correlation analysis was performed with the BRB array tools. Correlations were considered at the nominal 0.001 level of the univariate test.

Correlation Coefficient	Parametric *p*-Value	FDR	miRNA Name
0.953	<1 × 10^−7^	<1 × 10^−7^	mmu-miR-323-5p
0.888	9.17 × 10^−5^	0.00592	mmu-miR-1933-5p
0.888	9.17 × 10^−5^	0.00592	mmu-miR-1961
0.888	9.17 × 10^−5^	0.00592	mmu-miR-338-5p
0.888	9.17 × 10^−5^	0.00592	mmu-miR-504
0.884	0.0001922	0.0103	mcmv-miR-m107-1-5p
0.875	0.0003089	0.0125	mmu-miR-290-5p
0.874	0.0003089	0.0125	mmu-miR-708
0.857	0.0005971	0.0161	mmu-miR-181c
−0.857	0.0005971	0.0161	mmu-miR-2146
0.859	0.0005971	0.0161	mmu-miR-467h + mmu-miR-669d + mmu-miR-669l
0.857	0.0005971	0.0161	mmu-miR-669j
−0.843	0.0009695	0.0222	mmu-miR-93

## Data Availability

Raw proteomic and miRNA profiling data have been uploaded to the Store^db^ database (www.storedb.org, accessed on 30 January 2023). Study ID: https://www.storedb.org/?STOREDB:STUDY1176 (accessed on 30 January 2023), DOI: http://dx.doi.org/10.20348/STOREDB/1176 (accessed on 30 January 2023).

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
