# Peer review of "The miRNA Content of Bone Marrow-Derived Extracellular Vesicles Contributes to Protein Pathway Alterations Involved in Ionising Radiation-Induced Bystander Responses"

_ijms, 2023, doi:10.3390/ijms24108607_

Round 1

Reviewer 1 Report (Previous Reviewer 2)

Accept in present form

Author Response

Response to Reviewer 1

We thank the Reviewer for the thorough review process and for accepting the manuscript for publication.  

Reviewer 2 Report (Previous Reviewer 1)

The manuscript was successfully amended by the authors as advised.

However, I would strongly advise having some native English-speaking colleagues edit the entire manuscript.  

The revised draft of the manuscript is acceptable for publishing. 

There are many fundamentally incorrect sentences. For example, lines 1122–1125. A sentence will be grammatically incorrect when "although" is used as a subordinate conjunction in one clause and "but" is used as a coordinate conjunction in the other clause.

Lines 109-123; 644-648. Too complex to understand.

I would strongly advise having some native English-speaking colleagues edit the entire manuscript. 

Author Response

Response to Reviewer 2

We thank the Reviewer for the thorough revision of the manuscript. We corrected the indicated grammatical errors, we read through the whole manuscript, performed an English language editing and revised the sentences which were difficult to understand or were unclear.

We hope the actual, revised version of the manuscript is acceptable for publication. 

This manuscript is a resubmission of an earlier submission. The following is a list of the peer review reports and author responses from that submission.

Round 1

Reviewer 1 Report

Comments to authors: ijms-2222912

  1. In this study, miRNA analysis and the label-free proteomic platform were used to characterize the CBA mouse model prone to radiation-induced leukemia by analyzing the BM cells (BMC) and BM-EVs from irradiated mice. Indeed, comparative global proteome analysis is a solid experiment to identify the potential pathways and associated genes/proteins in response to IR stress. The experimental outline is well-designed. However, the data presentation is very sloppy and at some point, unsuitable for publication. In addition, some essential mandatory details about proteomic experiments are completely missing. For proteomics methodologies, authors followed an earlier report (J Proteome Res, 2020. 19(1): p. 337-345) which is very organized and easy to understand. I would suggest organizing their results similarly to that report. 

 Therefore, I strongly suggest a major revision. 

These are the major comments that should be taken care of before publication: 

  1. Unfortunately, I DO NOT see the info about the biological replications used for proteomic analysis. Please mention this point in the materials and methods section (Mass spectrometry (MS) sample preparation and measurement) and Figure 1. 
  2. It seems for LC-MS/MS analysis instrumentation set up authors completely followed the earlier published protocol (J Proteome Res, 2020. 19(1): p. 337-345) which is ok. For protein/peptide identification the PD (v 2.4), software with Sequest was used. Indeed, PD is one of the best software for label-free quantitative analysis. However, unfortunately, the authors did not utilize this software and adjunct pipeline (Minora algorithm) for quantitative analysis which is very surprising. Instead, it seems the authors performed a T-test which is not acceptable when multiple samples/groups and conditions were used for label-free proteomic analysis. The authors mentioned a reference where T-test was used [56]. It is important to note, in that reference paper isotope-labeled proteins/peptides were quantified. 
  3. Authors should analyze (label-free quantitative analysis) their proteomics datasets using PD 2.4. If authors do not have access to the quantitative pipeline of PD then the MaxQuant pipeline (https://www.maxquant.org/) should be used. I am sure authors are very well known that MaxQuant is a free pipeline for quantitative proteomic analysis. Here is the ref of PD vs MaxQuant pipeline (J. Proteome Res. 2021, 20, 7, 3497–3507).  
  4. Herein, 1.33-fold change was considered as a differentially abundant protein (DAP). While “1.5-fold” change is commonly used as a threshold in label-free quantitative analysis and considered as DAP. Therefore, the authors should justify their selection to this particular threshold (1.33-fold) as DAP. In support of their selection (fold change threshold), references should be given.
  5. Overall, the proteomics dataset was presented in a very poor manner. To clarify the quality of the overall proteomics dataset, A PCA analysis and heat map clustering for all conditions and proteins quantified should be provided. 
  6. Significantly up and down-regulated proteins should be presented by volcano plot analysis. 
  7. Tables 2 and 3, Figures 6-8 should be merged into a single complex figure. Instead of tables (Fig 6A, 7A, and 8A), these datasets could easily be presented by Venn diagram analysis.
  8. Figure 11: Authors should mention the justification for selecting 2 particular proteins for WB. 
  9. The conclusion should be revised and concise. Most of the sentences are very complex. For instance, the first sentence is compiled with 4 lines. Based on the miRNAs and proteomics analysis, a target protein set should be highlighted as potential biomarkers. 

Reviewer 2 Report

This study aims to characterize the changes in miRNA content of EVs and miRNA-regulated pathways and the proteome of BM cells treated with EVs from irradiated mice. By comparing the pathways regulated by differentially expressed miRNAs in EVs with altered protein pathways in BMCs treated with EVs from irradiated mice, the study aims to identify miRNA candidates involved in bystander signalling in EV-recipient BMCs. The study uses an in vivo model to investigate the role of EVs in mediating the radiation-induced bystander effect. The passage describes a study that characterizes extracellular vesicles (EVs) isolated from bone marrow (BM) of mice irradiated with ionizing radiation (IR) and compares the miRNA expression levels between the control and irradiated groups. The authors found that the miRNA cargo of BM-derived EVs showed a dose-dependent response to IR. They identified two miRNAs (mmu-miR-761 and mmu-miR-129-5p) that were increased in 0.1Gy EVs compared to EVs from controls, while 17 miRNAs were differentially expressed in 3Gy EVs. Pathway analysis revealed that the altered pathways were linked to cancer, signal transduction, and cellular processes. Clustering analysis showed clear up- and down-regulated clusters of miRNAs which included the miRNAs identified using Diana tools software. The authors also identified 13 miRNAs that showed a strong dose-response with correlation coefficients between 0.95 and 0.84.

The abstract should not be divided into paragraphs and it is too lengthy.

In terms of related studies, there has been increasing interest in the role of EVs in intercellular communication and their potential as biomarkers and therapeutic agents. Several studies have investigated the effects of radiation on EVs and their cargo, including miRNAs, proteins, and lipids, and their role in mediating radiation-induced bystander effects. Some studies have also investigated the effects of EVs derived from irradiated cells on non-irradiated cells and the potential mechanisms involved. My question is: What are some potential implications of the findings from this study for understanding tissue responses to radiation exposure and estimating long-term risks after radiation exposure?

What were the study's main findings on differentially expressed miRNAs in BM-derived EVs from mice irradiated with low or high-dose IR, and how were these findings linked to alterations in the biological processes of EV acceptor cells?

What are the main findings of the study regarding the differences in protein pathways between BMCs directly irradiated and BMCs treated with EVs from irradiated mice?

I congratulate the authors on a well-structured and accomplished article.